# Characterization of the Bacteriome of *Culicoides reevesi* from Chihuahua, Northern Mexico: Symbiotic and Pathogenic Associations

**DOI:** 10.3390/insects17010052

**Published:** 2026-01-01

**Authors:** Rodolfo González-Peña, David Orlando Hidalgo-Martínez, Stephanie V. Laredo-Tiscareño, Herón Huerta, Erick de Jesús de Luna-Santillana, Jaime R. Adame-Gallegos, Carlos A. Rodríguez-Alarcón, Ezequiel Rubio-Tabares, Julián E. García-Rejón, Zilia Y. Muñoz-Ramírez, Chandra Tangudu, Javier A. Garza-Hernández

**Affiliations:** 1Laboratorio de Arbovirología, Centro de Investigaciones Regionales “Dr. Hideyo Noguchi”, Universidad Autónoma de Yucatán, Mérida 97000, Yucatán, Mexico; fitho.gleez@outlook.com (R.G.-P.); julian.garcia@correo.uady.mx (J.E.G.-R.); 2Instituto de Ciencias Biomédicas, Universidad Autónoma de Ciudad Juárez, Ciudad Juárez 32310, Chihuahua, Mexico; dhidalgom2500@alumno.ipn.mx (D.O.H.-M.); viridiana.laredo@gmail.com (S.V.L.-T.); carrodri@uacj.mx (C.A.R.-A.); erubio@uacj.mx (E.R.-T.); 3Laboratorio Medicina de la Conservación, Centro de Biotecnología Genómica del Instituto Politécnico Nacional, Reynosa 88710, Tamaulipas, Mexico; 4Laboratorio de Entomología, Instituto de Diagnóstico y Referencia Epidemiológicos, Delegación Álvaro Obregón, Ciudad de México 01480, Mexico; cerato_2000@yahoo.com; 5Facultad de Ciencias Químicas, Universidad Autónoma de Chihuahua, Chihuahua 31125, Chihuahua, Mexico; jadame@uach.mx (J.R.A.-G.); zramirez@uach.mx (Z.Y.M.-R.); 6Panthera Vax, LLC., Ames, IA 96701, USA; ctangudu@pantheravax.com

**Keywords:** *Culicoides reevesi*, bacteriome, metagenomics, 16s rRNA, Chihuahua, Mexico

## Abstract

Biting midges are very small insects that can transmit diseases to animals and humans, yet we know little about the bacteria that live inside them. In this study, we examined the bacteria associated with *Culicoides reevesi*, a species found in northern Mexico that has not been well studied. We collected groups of adult midges and analyzed their bacteria using genomic and bioinformatic methods. We found that some of the bacteria may help in the metabolic process of insects, while others are known to cause disease in animals or humans. The bacteria also showed signs of being involved in important processes such as energy use, production of essential nutrients, and communication between microbes. Interestingly, even though all insects were collected from the same place and time, the types of bacteria varied between groups, suggesting that local conditions or differences among the insects themselves may play a role. This is the first study to describe the bacteria of *C. reevesi*. These findings can support better understandings of disease risks and may help to future strategies for controlling biting midges.

## 1. Introduction

The microbiome, defined as the community of microorganisms residing on or within a host, plays a crucial role in the physiology of arthropod vectors, including *Culicoides* biting midges. This community, comprising bacteria, viruses, protozoa, and fungi, can be transmitted both horizontally and vertically, thereby influencing host physiology and phenotype [1,2,3]. In *Culicoides* and other medically significant arthropods, microbiome acquisition and composition are shaped by a range of biotic and abiotic factors, including host genetics and environmental conditions [4,5,6,7,8]. Consequently, the microbiome can vary significantly across individuals, developmental stages [9,10,11], species, and geographic regions [4,12], contributing to differences in host phenotypes [13].

Microbiomes in insect vectors affect pathogen transmission both directly and indirectly, often through symbiotic relationships that can be exploited for disease control [14,15]. Among *Culicoides* species, certain midges are primary vectors of zoonotic pathogens, including viruses, bacteria, protozoa, and nematodes [16,17,18,19]. In the Americas, *Culicoides* transmit diseases such as bluetongue virus (BTV), epizootic hemorrhagic disease (EHD), and vesicular stomatitis virus (VSV), affecting livestock and wildlife [20,21]. *Culicoides paraensis* has also been associated with recent Oropouche virus outbreaks in South America (Brazil, Peru, Colombia, Cuba, and others) [22].

Research on *Culicoides* microbiota has predominantly focused on midgut bacteria, which are often identified only at the genus level [4,11,23,24]. While this provides a baseline understanding of microbial communities, it offers limited insights into the functional roles of these microbes in shaping vector competence. Comprehensive profiling of *Culicoides* microbiota, including functional predictions, could reveal the ecological roles of key taxa and uncover novel strategies to mitigate pathogen transmission. However, studies on mosquitoes suggest that midgut bacteria can similarly influence the vector competence of *Culicoides* in transmitting pathogens [25]. Studies have detected endosymbionts like *Wolbachia*, *Cardinium*, and *Rickettsia* in *Culicoides* across Japan, Australia, and Europe, with potential implications for vector competence [26,27,28,29,30,31]. Antibiotic treatments in *Culicoides nubeculosus* and *Culicoides sonorensis* altered midgut microbial communities, resulting in increased Schmallenberg virus (SBV) infection rates, suggesting a protective role of the midgut microbiota in pathogen inhibition [12]. Additionally, symbionts present in *Culicoides* offer promising opportunities for genetic manipulation to reduce vector competence, as demonstrated in mosquitoes [32].

Studies in mosquitoes have demonstrated that gut microbiota diversity plays a critical role in modulating immune responses and influencing pathogen susceptibility [33]. This suggests that similar mechanisms may operate in *Culicoides* species, highlighting the importance of understanding their microbiome. *Culicoides reevesi* has recently emerged as a species of interest in Mexico due to its aggressive biting behavior towards humans and animals and its potential as a vector of arboviruses [34]. Seven novel viruses associated with *C. reevesi* in Chihuahua, Mexico, have been previously identified, spanning five viral families, Nodaviridae, Partitiviridae, Solemoviridae, Tombusviridae, and Totiviridae, as well as one unclassified virus [35]. Despite its ecological and epidemiological significance, little is known about the microbiome of *C. reevesi*.

Based on this background, we hypothesize that the microbiome of *C. reevesi* plays a dual ecological role, influencing both host biology and pathogen transmission. Symbiotic bacteria may contribute to host physiology and modulate vector competence, whereas potentially pathogenic taxa may represent environmental or opportunistic associations that affect host fitness and disease dynamics. Understanding the coexistence of these microbial groups will provide insights into how the *C. reevesi* microbiome shapes its vectorial capacity and may inform future microbiome-based control strategies. Therefore, the objective of this study is to provide the first comprehensive metagenomic sequencing analysis of the bacterial diversity within *C. reevesi* from Chihuahua, Mexico, and provide valuable insights into its microbiome and potential implications for vector biology, public health, and disease control strategies.

## 2. Materials and Methods

### 2.1. Insect Collection and Morphological Identification

In July 2023, adult females (n = 125) of *Culicoides reevesi* were collected by human landing catches in Buenaventura, Chihuahua, near the Santa María River (29°50′37″ N, 107°28′19″ O; 1553 MASL), during routine mosquito surveillance activities (Figure 1). Sampling was conducted on a single day between 16 and 20 h, coinciding with high biting activity. The specimens were preserved in 1.5 mL polypropylene tubes containing 96% ethanol and stored at −20 °C until further processing, which included morphological identification and gDNA extraction for 16S rRNA amplicon sequencing analysis. Identification was performed according to the taxonomic guidelines outlined by González-Peña et al. (2025) [34], focusing on diagnostic characteristics such as wing patterns, antennal segmentation, and other morphological traits distinctive to *C. reevesi*.

### 2.2. DNA Isolation, PCR, and Sequencing

Specimens were cleaned individually with nuclease-free water, grouped into 5 pools of 25 individuals, and placed in 1.5-mL polypropylene tubes. DNA was isolated using DNeasy Blood and Tissue Kit (Qiagen^©^, Redwood City, CA, USA. Cat. no. 69504, Germantown, MD, USA). A volume of 180 µL of ATL lysis buffer and 20 µL of Proteinase K (20 mg/mL) was added to each tube for specimen maceration with a sterile pestle. Samples were then incubated overnight at 56 °C in an Eppendorf thermomixer shaker (Eppendorf, Hamburg, Germany). Subsequently, DNA extraction was completed following the manufacturer’s instructions, including sequential washes with AL, AW1, and AW2 buffers. The isolated DNA was stored in PCR microtubes in 100 µL of AE dilution buffer. Of the five pools processed, only four reached the quality standards required for downstream sequencing.

Sequencing was carried out to according to the Illumina 16S Metagenomic Sequencing Library Preparation protocol (Part #15044223 Rev. B, San Diego, CA, USA). Specific primers were used to amplify the V3–V4 regions of the 16S rRNA gene from extracted DNA (Forward: 5′-TCGTCGGCAGCGTCAGATGTGTATAAGAGACAGCCTACGGGNGGCWGCAG-3′; Reverse: 5′-GTCTCGTGGGCTCGGAGATGTGTATAAGAGACAGGACTACHVGGGTATCTAATCC-3′). The PCR mix (25 μL) included 5 ng/μL gDNA, 5 μL of 1 μM forward and reverse primers, and 12.5 μL of 2× KAPA HiFi HotStart Ready Mix. The amplification protocol consisted of an initial denaturation at 95 °C for 3 min, followed by 25 cycles of denaturation at 95 °C for 30 s, annealing at 55 °C for 30 s, extension at 72 °C for 30 s, and a final elongation at 72 °C for 5 min. Then, the PCR products were purified with AMPure XP beads (Beckman Coulter Life Sciences, Brea, CA, USA) to remove primer residues [36]. A second PCR was performed using the Nextera XT Index Kit, following Illumina’s protocol (2013) to generate paired-end reads (2 × 150 bp). Raw sequence data have been deposited in the NCBI Sequence Read Archive (SRA) under BioProject accession number PRJNA1332119.

### 2.3. Sequence Processing, Taxonomic Classification, and Diversity Analysis with Galaxy

Paired-end sequencing reads in FASTQ format were generated with the Illumina platform and analyzed through the Galaxy Tool Shed platform (https://usegalaxy.eu), which integrates the QIIME2 (v2025.4) microbiome analysis framework. Sequence processing was performed with the publicly available workflow “QIIME2 Id: Demultiplexed data (paired-end, release v0.3)”, shared by user iwc. After importing and demultiplexing the data, sequence quality was assessed using QIIME2 View, which revealed that the reverse reads were of insufficient quality and read length for reliable downstream processing. Consequently, only the forward reads were retained for analysis. Amplicon sequence variants (ASVs) were inferred using QIIME2 DADA2 Denoise-single tool within Galaxy, using a trim length parameter of 140 bp. This method performs quality filtering, error correction, and chimera removal to generate high-resolution ASVs [37]. Subsequently, taxonomic classification was performed using the classify-sklearn method in QIIME2 [38], with a pre-trained Silva classifier (99% similarity threshold) as the reference database [39].

Taxonomic assignments were visualized with the QIIME 2 taxa bar plot tool and explored in QIIME 2 View. Stacked bar plots for Phylum, Class, Order, Family, and Genus were generated in Python (v3.11) using pandas (v2.2.2) and matplotlib (v3.9.0). Counts were converted to relative abundances per pool; taxa below 1% relative abundance (per plot) were grouped as Others for visualization. For each pool and rank, Unclassified was defined as the proportion of ASVs or reads lacking a valid rank label (e.g., unclassified/unknown/norank/NA/uncultured/sp.). Results are reported per pool and rank.

For diversity analyses, the alpha diversity (Observed, Chao1, Shannon, Simpson [1–D]) was computed from the rarefied feature table (254 reads per pool). Beta diversity was assessed with Bray–Curtis and visualized via PCoA (PCo1–PCo2; variance explained on axes). Rarefaction curves of expected Observed ASVs (≤254 reads) were derived analytically from the raw table.

### 2.4. Detection of Pathogenic and Symbiotic Bacteria Using CZ ID

Raw sequencing data were processed using the open-source CZ ID platform (Chan Zuckerberg ID, https://czid.org), which assigns taxonomic identities based on alignments to the NCBI nucleotide (NT) and non-redundant protein (NR) databases. For pathogen detection, analysis was restricted to the NT database, which includes both coding and non-coding regions—such as rRNA genes—crucial for accurate bacterial identification.

To identify clinically relevant bacterial taxa, the results were filtered by selecting the “Bacteria” category, applying the “Known Pathogens” tag, and limiting the taxonomic resolution to the species level. Bacterial abundance was estimated using the NT rPM (reads per million) metric, which normalizes for differences in sequencing depth across samples.

A complementary analysis was performed to explore symbiotic bacterial genera commonly associated with blood-feeding arthropods. Taxa were grouped at the genus level, and both samples and genera were sorted alphabetically to facilitate comparison. To minimize the influence of highly abundant taxa and enhance data visualization, NT rPM values were log-transformed using the formula log_10_ (rPM + 1). Heatmaps were generated in Python (v3.11) using the libraries pandas (v2.2.2), seaborn (v0.13.2), and matplotlib (v3.9.0) to illustrate the distribution of pathogenic species and key symbiotic genera across samples.

### 2.5. Functional Prediction and Pathway Classification (PICRUSt2)

Amplicon sequence variants (ASVs) obtained after quality control and denoising (Denoise single; trimmed length 140 bp) were used as input for functional prediction with PICRUSt2 (Galaxy Version 2.5.3 + galaxy0) [40] on Galaxy EU. The workflow comprised: (i) phylogenetic placement of ASVs into a reference tree with SEPP (default settings), (ii) hidden-state prediction of gene family copy numbers using CASTOR, and (iii) inference of KEGG Ortholog (KO) and Enzyme Commission (EC) profiles. Predicted gene families were then collapsed to MetaCyc pathways using the PICRUSt2 “pathway_map” step with the default MinPath implementation.

To reduce spurious predictions, (a) ASVs with NSTI > 2.0 were removed, (b) the default 16S copy-number normalization was applied, and (c) pathways with non-zero abundance in at least one sample were retained. Per-sample abundances were normalized to relative abundance (sum to 1) before between-sample summaries. For reporting, pathways were grouped into higher-level MetaCyc classes (e.g., Biosynthesis, Generation of Precursor Metabolites and Energy, Degradation/Utilization/Assimilation). The Top-25 pathways were defined by cohort-level total predicted pathway abundance (i.e., the sum of unstratified predicted copies across all samples for each pathway) and, together with class summaries, were visualized in GraphPad Prism (v.10) without additional transformations (Table 1). No inferential statistics were applied; results are presented descriptively.

## 3. Results

### 3.1. Sequencing Overview

A total of 829,756 raw paired-end reads were generated from four pooled samples, each consisting of 25 *C. reevesi* individuals. After quality control filtering (average Phred quality score ≥ 30, corresponding to a base call accuracy of 99.9%), 39,213 high-quality reads were retained. The number of high-quality reads varied across samples: sample 1EL yielded 11,513 reads, sample 2EL produced 2854 reads, sample 3EL generated 17,460 reads, and sample 5EL retained 7386 reads.

### 3.2. Microbial Composition of Culicoides reevesi

The global bacteriome of *C. reevesi* was dominated by Pseudomonadota (66%), followed by Actinomycetota (28%), Bacillota (5%), and Bacteroidota (1%), while less abundant phyla were grouped as others (Figure 2, Panel A). These results are consistent with microbial profiles reported in other blood-feeding arthropods, where Pseudomonadota plays essential roles in nutrient metabolism, host interactions, and potential vector competence. Despite this global pattern, pool-specific variation was evident (Figure 2, Panel B). Pseudomonadota predominated in pools 1EL, 2EL, and 5EL (93%, 80%, and 97%, respectively), whereas Actinomycetota was the most abundant in 3EL (52%), slightly surpassing Pseudomonadota (45%). Bacteroidota was notably enriched in 2EL (19%) relative to the other pools. Rank-resolved barplots (Phylum to Genus) are shown in Appendix A, confirming dominant Pseudomonadota and Actinomycetota and highlighting *Methylorubrum/Methylobacterium*, *Cutibacterium*, and *Staphylococcus/Streptococcus* as leading genera, with *Spiroplasma*, *Apibacter*, and “*Candidatus Cardinium*” at lower, variable levels. This variation in microbial composition across pools may reflect differences in environmental exposure, diet, or host-related factors. The observed microbial diversity and dominance of specific bacterial taxa could influence the role of *C. reevesi* as a potential vector, either by interacting with pathogens or contributing to host fitness.

The fraction of Unclassified assignments was low at the Phylum level and increased progressively toward the Genus level, as quantified by ASVs and reads (Appendix A; Appendix A). This pattern was consistent whether measured by features (ASVs) or by total reads, indicating that a non-trivial portion of the community remains unresolved at lower ranks. These findings point to underrepresented taxa in available reference databases and suggest that some ecological patterns may be underestimated due to unresolved diversity.

### 3.3. Diversity Analyses

Rarefaction curves of expected Observed ASVs showed rapid richness accumulation at low depths with partial stabilization near 254 reads (Appendix A). Using the rarefied table (depth = 254), alpha diversity indicated moderate between-pool differences (Observed, Chao1, Shannon, Simpson; Appendix A). Bray–Curtis dissimilarities visualized by PCoA, with pools separated despite the shared site/date, with PCo1 = 40.84% and PCo2 = 34.80% (75.64% cumulative; Appendix A; Appendix A).

### 3.4. Pathogenic and Symbiotic Bacteria Detected in Culicoides reevesi

The 16S rRNA amplicon analysis of *Culicoides reevesi* revealed the coexistence of both pathogenic and symbiotic bacteria across pooled samples of 25 individuals (Figure 3). Within the pathogenic community (Figure 3, Panel A), species of clinical and veterinary concern were identified, including *Mycobacterium avium*, *Enterococcus faecalis*, *Escherichia coli*, *Nocardia brasiliensis*, and *Acinetobacter baumannii*. Notably, *E. faecalis* and *M. avium* exhibited the highest relative abundance, with considerable variation observed among different pools.

In contrast, the symbiotic fraction of the microbiota (Figure 3, Panel B) was characterized by genera commonly associated with arthropod vectors, such as *Spiroplasma*, *Cardinium*, *Rickettsia*, and *Asaia*. Other genera detected were *Lactobacillus*, *Stenotrophomonas*, and *Pseudomonas*, representing a mix of beneficial, commensal, and opportunistic bacteria. This composition reflects a diverse microbial community inhabiting *C. reevesi*.

### 3.5. Functional Prediction and Pathway Classification (PICRUSt2) in C. reevesi

Using PICRUSt2, predicted functional profiles were dominated—at the class level—by Biosynthesis (50.49%), followed by Generation of Precursor Metabolites and Energy (34.70%) and Degradation/Utilization/Assimilation (13.56%) (Figure 4). Within the top-25 predicted MetaCyc pathways (ranked by cohort-level total predicted pathway abundance), central-carbon processes—aerobic respiration (cytochrome c), the tricarboxylic acid (TCA) cycle, fatty-acid β-oxidation, and glycolysis—predominated, indicating active energy metabolism in the community. Key biosynthetic routes were also prominent, including phospholipid and peptidoglycan biosynthesis, heme and folate biosynthesis, and nucleotide metabolism (Table 1). A siderophore pathway (aerobactin biosynthesis) was detected at lower abundance, suggesting potential iron-acquisition capacity (Table 1).

## 4. Discussion

The bacteriome analysis of *C. reevesi* provides valuable insight into the diversity, composition, and potential roles of its associated bacterial communities. These microbes may influence host metabolism, immune function, and vector competence. As an exploratory 16S rRNA amplicon-based study, the goal was to generate baseline data that can guide future research on bacteriome-mediated interactions in this species. Their ecological and epidemiological significance underscores the identification of both symbiotic and pathogenic bacteria, while the differences observed across samples indicate a strong influence of environmental and host-related factors.

Pseudomonadota was identified as the dominant phylum, comprising 66% of the bacteriome, followed by Actinomycetota (28%), Bacillota (5%), and Bacteroidota (1%), with other phyla contributing less than 1% (grouped as Others). This phylum-level profile is consistent with those reported in *C. sonorensis* and *C. stellifer* [11,41], suggesting a conserved bacterial community structure across *Culicoides* species. Within Pseudomonadota, the prevalence of genera such as *Enterobacter*, alongside *Mycobacterium* and *Bacillus* in Actinomycetota and Bacillota, respectively, mirrors patterns observed in other hematophagous insects [6,24]. These taxa have been implicated in immune modulation and interactions with pathogens, potentially influencing the ability of vectors to acquire and transmit disease.

Importantly, the presence of symbiotic bacteria, such as *Enterobacter*, *Pseudomonas*, *Spiroplasma*, *Asaia*, and *Cardinium*, opens potential avenues for bacteriome-based vector control strategies. Endosymbionts, such as *Wolbachia* and *Cardinium*, have been demonstrated to manipulate host reproduction and reduce pathogen transmission in other Diptera [27,31]. In *Culicoides*, Torix-group *Rickettsia* have been detected within the ovaries of *Culicoides newsteadii* and recognized as vertically transmitted, obligate endosymbionts that occur at high frequencies across multiple species [31]. Therefore, the genome detection of *Rickettsia* in *C. reevesi* most likely reflects a stable symbiotic association with potential implications for host biology and vector competence. The application of paratransgenesis leveraging these symbionts could offer a promising tool for limiting vector competence in *C. reevesi* and related species [6,42].

The detection of potentially harmful bacteria—including *Mycobacterium avium*, *Vibrio parahaemolyticus*, *Escherichia coli*, *Enterococcus faecalis*, and *Nocardia brasiliensis* (Figure 4)—raises important ecological and public health concerns. This aligns with earlier culture-independent studies of *Culicoides* midguts [11]. Despite all samples being collected from the same site on the same day, the heat maps show clear differences in pathogen composition and abundance between pools. This suggests microhabitat variability or bacterial input from host sources. Similar small-scale spatial effects on midge microbiomes have been reported in European *Culicoides* [12]. Many of these bacteria are commonly found in feces, soil, or decaying organic matter, which likely explains their presence. The frequent detection of *E. coli* and *E. faecalis* across pools points to fecal or organic contamination in larval habitats [4,43], consistent with their use as standard indicators of fecal pollution in water systems [44].

Lower-abundance bacteria such as *Serratia marcescens* and *Streptococcus gordonii*, both, which are known as commensal or opportunistic microbes, may play a role in bacteriome function through antimicrobial production, competition, or niche modification. For example, *S. marcescens* strains isolated from mosquitoes carry genes for antimicrobial compounds and can suppress other microbes [45]. These interactions could influence pathogen colonization and vector competence. Disrupting midge gut bacteria with antibiotics has been shown to increase Schmallenberg virus infection rates in *C. nubeculosus* [24], mirroring earlier findings on how gut microbiota affect arbovirus spread [11]. From a public health and veterinary perspective, the presence of zoonotic or hospital-associated bacteria like *M. avium*, *Acinetobacter baumannii*, and *Salmonella enterica* is notable. Recent research suggests *A. baumannii* can persist in insect guts and even boost host immunity [46], highlighting the potential for midges to act as accidental reservoirs or mechanical carriers between wildlife, livestock, and humans.

In addition to taxonomic composition, functional predictions using PICRUSt2 revealed that the bacterial community of *C. reevesi* is dominated by pathways related to central carbon metabolism, cell envelope biosynthesis, and vitamin/cofactor metabolism. The detection of siderophore biosynthesis and quorum-sensing pathways is particularly relevant, as these functions can enhance bacterial competitiveness within the insect host and potentially modulate pathogen colonization [47,48]. Such functional insights complement the taxonomic findings, providing a more comprehensive picture of the ecological potential of these bacterial communities. Unlike Banerjee et al. (2023) [9], who restricted their analysis to the gut microbiome of *C. peregrinus*, the whole-body pooling approach captured bacterial taxa and functions associated with both internal and external niches, broadening the ecological context of vector–microbe interactions in *Culicoides*.

Some methodological limitations should be considered. Low-biomass 16S studies are prone to contamination from lab reagents; genera like *Acinetobacter* and *Pseudomonas* often appear as kit contaminants [49]. Pooling samples also hides individual variation. Genus-level interpretation was adopted because the 16S V4 amplicon (515F/806R) and SILVA-based workflows are most reliable at the genus level for community profiling [39]. Consistent with this constraint, the *Unclassified* fraction was low at the phylum level and increased toward genus (by ASVs and reads), indicating limited reference resolution at finer ranks [38,39]. After standardizing depth, alpha-diversity (Observed, Chao1, Shannon, Simpson) showed moderate between-pool differences, and beta-diversity based on Bray–Curtis principal coordinates analysis (PCoA) separated pools despite a shared site/date, reflecting shifts in relative abundances rather than presence/absence alone; this is consistent with diversity analyses reported for *Culicoides* microbiomes [4,12]. Future studies should combine long-term and individual sequencing with strict negative controls and environmental sampling to distinguish true symbionts from transient or contaminant bacteria.

Finally, the coexistence of symbiotic and potentially pathogenic bacteria within the *C. reevesi* microbiome reflects a complex ecological network that may influence host physiology and vector competence. Symbiotic taxa such as *Asaia*, *Cardinium*, and *Rickettsia* could contribute to host physiological balance and pathogen exclusion through immune modulation, resource competition, or interference with pathogen colonization. In contrast, the detection of opportunistic or pathogenic species such as *E. coli* and *Mycobacterium* spp. may indicate temporary environmental acquisition or opportunistic infections that arise under altered physiological conditions. The connections between these microbial groups likely determine the structural and functional *C. reevesi* bacteriome, underscoring the importance of microbiome composition in modulating vectorial capacity and pathogen transmission dynamics.

## 5. Conclusions

This exploratory bacteriome analysis of *C. reevesi* revealed a community dominated by Pseudomonadota, with additional representation of Actinomycetota, Bacillota, and Bacteroidota. The detection of symbiotic taxa such as *Asaia* and *Cardinium*, alongside potentially pathogenic bacteria including *Escherichia coli* and *Mycobacterium avium*, highlights the dual role of the bacteriome in both maintaining host fitness and influencing pathogen transmission potential. Functional predictions indicated metabolic versatility, with pathways related to aerobic respiration, amino acid biosynthesis, and quorum sensing, suggesting bacterial contributions to host–microbe interactions.

Our findings provide the first taxonomic and functional baseline of the *C. reevesi* bacteriome. While limited by sequencing depth, pooling strategy, and the use of 16S rRNA predictive tools, this study underscores the ecological and epidemiological significance of midge-associated bacteria. Future research should prioritize longitudinal and individual-based sampling, coupled with shotgun metagenomics, to clarify the ecological roles of key bacterial taxa and their influence on vector competence. Such studies will support the integration of microbiome analyses into entomological surveillance and the development of microbiome-based strategies for vector control.

## Figures and Tables

**Figure 1 insects-17-00052-f001:**
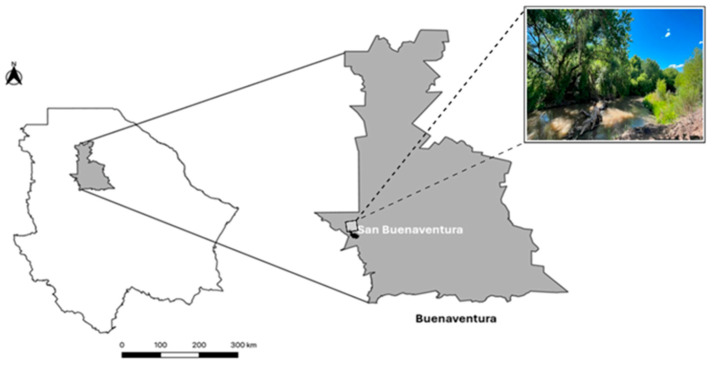
Geographic location of the collection site in Chihuahua, México. The map shows the state of Chihuahua, with the Buenaventura municipality shaded in dark gray, and the city of San Buenaventura highlighted in a darker shade.

**Figure 2 insects-17-00052-f002:**
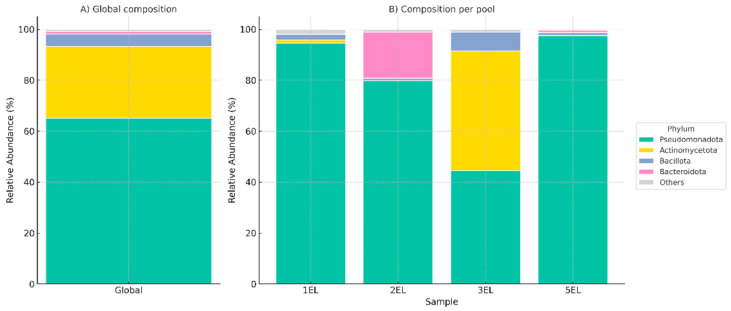
Bacteriome composition of *Culicoides reevesi*. (**A**) Global relative abundance of bacterial phyla across all pooled samples, dominated by
*Pseudomonadota*, followed by
*Actinomycetota*,
*Bacillota*, and
*Bacteroidota*, with less abundant phyla grouped as
*Others*. (**B**) Relative abundance at the phylum level across individual pools (1EL, 2EL, 3EL, and 5EL), highlighting pool-specific variation.

**Figure 3 insects-17-00052-f003:**
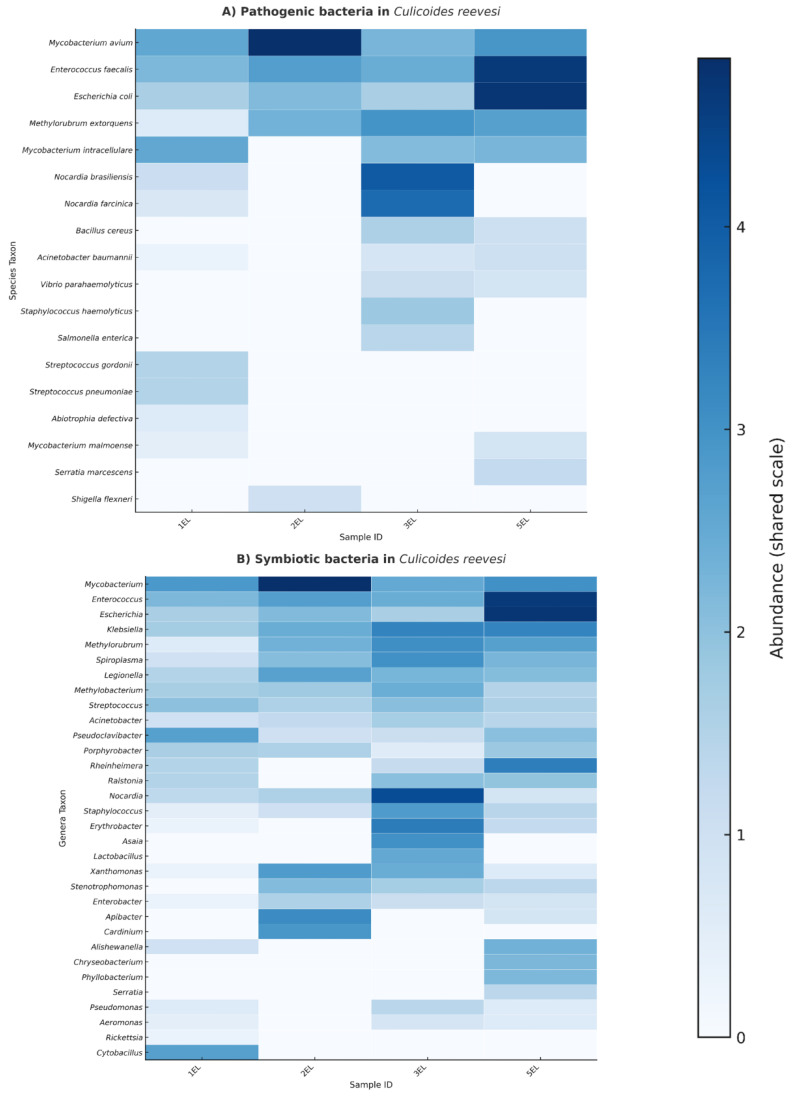
Pathogenic and symbiotic bacteria associated with *C. reevesi*. (**A**) Heatmap showing the relative abundance (log-transformed values) of potentially pathogenic bacterial species detected in different pooled samples (1EL, 2EL, 3EL, and 5EL). (**B**) Heatmap illustrating the distribution of symbiotic bacterial genera across the same pools. Color intensity reflects relative abundance, with darker shades indicating higher abundance.

**Figure 4 insects-17-00052-f004:**
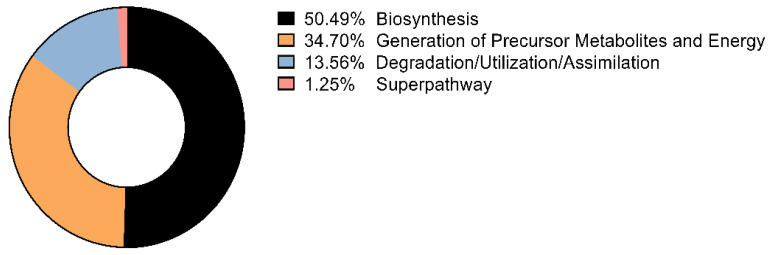
Functional pathway categories predicted by PICRUSt2 in *Culicoides reevesi*. Relative predicted pathway composition grouped into MetaCyc classes: Biosynthesis (50.49%), Generation of Precursor Metabolites and Energy (34.70%), Degradation/Utilization/Assimilation (13.56%), and Superpathway (1.25%). Percentages are cohort-level (sum of predicted copies across all samples within each class, normalized to 100%). Visualization performed without additional transformations.

**Table 1 insects-17-00052-t001:** Top 25—Total predicted metabolic pathways abundances in *C. reevesi* bacteriome (PICRUSt2, unstratified).

Class	Pathway ID	Pathway	Total Predicted Pathway Abundances *
Generation of Precursor Metabolites and Energy	PWY-3781	Aerobic respiration I (cytochrome c)	78,594
Degradation/Utilization/Assimilation	P221-PWY	Octane oxidation	41,276
Generation of Precursor Metabolites and Energy	TCA	TCA cycle 1 (prokaryotic)	35,786
Degradation/Utilization/Assimilation	FAO-PWY	Fatty acid β-oxidation I	35,048
Generation of Precursor Metabolites and Energy	PWY-7111	Pyruvate fermentation to isobutanol	34,927
Biosynthesis	PHOSLIPSYN-PWY	Superpathway of phospholipid biosynthesis III (*E. coli*)	32,452
Biosynthesis	POLYISOPRENSYN-PWY	Polyisoprenoid biosynthesis (*E. coli*)	30,715
Biosynthesis	PWY4FS-8	Phosphatidylglycerol biosynthesis II	30,457
Biosynthesis	PWY4FS-7	Phosphatidylglycerol biosynthesis I	30,457
Generation of Precursor Metabolites and Energy	GLYCOLYSIS	Glycolysis	29,916
Biosynthesis	DAPLYSINESYN-PWY	L-lysine biosynthesis I	28,957
Generation of Precursor Metabolites and Energy	ANAGLYCOLYSIS-PWY	Glycolysis III	28,551
Biosynthesis	PEPTIDOGLYCANSYN-PWY	Peptidoglycan biosynthesis I	26,603
Biosynthesis	HEMESYN2-PWY	Heme *b* biosynthesis II (oxygen-independent)	26,046
Biosynthesis	TRNA-CHARGING-PWY	tRNA charging	22,327
Biosynthesis	HEME-BIOSYNTHESIS-II	Heme *b* biosynthesis II (oxygen-independent)	21,376
Biosynthesis	1CMET2-PWY	Folate Transformations III (*E. coli*)	20,943
Biosynthesis	BIOTIN-BIOSYNTHESIS-PWY	Biotin biosynthesis I	14,843
Biosynthesis	DENOVOPURINE2-PWY	Superpathway of purine nucleotides de novo biosynthesis II	14,095
Superpathway	ALL-CHORISMATE-PWY	Superpathway of chorismate metabolism	7532
Degradation/Utilization/Assimilation	PROTOCATECHUATE-ORTHO-CLEAVAGE-PWY	Protocatechuate degradation II (ortho-cleavage pathway)	3156
Biosynthesis	TEICHOICACID-PWY	Poly (glycerol phosphate) wall teichoic acid biosynthesis	3012
Degradation/Utilization/Assimilation	ORNARGDEG-PWY	Superpathway of L-arginine and L-ornithine degradation	884
Degradation/Utilization/Assimilation	CATECHOL-ORTHO-CLEAVAGE-PWY	Catechol degradation to β-ketoadipate	829
Biosynthesis	AEROBACTINSYN-PWY	Aerobactin biosynthesis	124

* Values are the sum of predicted copies for each MetaCyc pathway across all pools.

## Data Availability

The original contributions presented in this study are included in the article/Appendix A. Further inquiries can be directed to the corresponding author.

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
