# Peer review of "Characterization of the Bacteriome of Culicoides reevesi from Chihuahua, Northern Mexico: Symbiotic and Pathogenic Associations"

_insects, 2026, doi:10.3390/insects17010052_

Round 1
Reviewer 1 Report
Comments and Suggestions for Authors
The dataset presented in this manuscript appears rather limited and may not provide sufficient information to support strong or novel conclusions. The study’s level of innovation is relatively modest. Moreover, functional prediction based on amplicon sequencing has inherent limitations, and the lack of experimental validation further weakens the reliability of the findings. In light of these issues, I do not recommend the manuscript for publication in its current form.
Author Response
Dear Reviewer,
Thank you for the valuable feedback. We would like to clarify that this manuscript is submitted as a Communication, which, according to the journal Insects (MDPI) guidelines, are short articles presenting groundbreaking preliminary results or significant findings that form part of a larger, ongoing study.
The other reviewers provided constructive comments that have been fully addressed, and we believe these revisions have substantially improved the quality and clarity of the manuscript.
Thank you a lot in advance,
Best regards
Reviewer 2 Report
Comments and Suggestions for Authors
Overall a very straightforward and nicely written piece. It is important baseline for culicoides revesii and will likely be useful for future microbiome work in the species.
Major corrections
NEEDS a data availability section. You must upload your microbiome data to genbank or elsewhere.
Figure 3: aesthetics alterations needed. Make sure your font sizes are all the same and don’t accidentally squash your images. Some of the text looks compressed. It may help to plot them as subfigures? (like this: https://www.tutorialspoint.com/combining-two-heatmaps-in-seaborn). Also, The line about colour intensity does not hold for both plots (yellow is not a dark colour), i’d consider using the same colour palette for both so this is consistent.
Minor corrections
77-78: On first read through this sounded like you're calling culicoides mosquitoes! Consider rewording for clarity
312: Possibly relevant thing, Rickettsia has been found in the ovaries of Culicoides newsteadii and is a vertically transmitted obligate endosymbiont https://doi.org/10.1111/1462-2920.13887 . It’s unlikely that it is transient or a contaminant.
Author Response
Thank you for your valuable comments, which have substantially improved the content and clarity of the manuscript.
Author's Reply to the Review Report (Reviewer 2)
Major corrections
Comment 1: NEEDS a data availability section. You must upload your microbiome data to genbank or elsewhere.
Answer 1: Thank you for the comment. The sequencing project has already been approved and registered in GenBank. The data have been deposited in the NCBI Sequence Read Archive (SRA) under BioProject accession number PRJNA1332119 (SubmissionID: SUB15653780).
Comment 2: Figure 3: aesthetics alterations needed. Make sure your font sizes are all the same and don’t accidentally squash your images. Some of the text looks compressed. It may help to plot them as subfigures? (like this: https://www.tutorialspoint.com/combining-two-heatmaps-in-seaborn). Also, The line about colour intensity does not hold for both plots (yellow is not a dark colour), i’d consider using the same colour palette for both so this is consistent.
Answer 2: Thank you for the suggestion. We have re-created the Figure 3 following your recommendations: standardized all font sizes, avoided any image compression, arranged the panels as subfigures for clarity, unified the color palette across both plots, and revised the caption/legend to ensure the statement about color intensity is consistent. The updated figure now presents the results more clearly.
Minor corrections
Comment 3: 77-78: On first read through this sounded like you're calling culicoides mosquitoes! Consider rewording for clarity.
Answer 3: Rewording was already made. The revised sentence is no longer confusing.
Comment 3: 312: Possibly relevant thing, Rickettsia has been found in the ovaries of Culicoides newsteadii and is a vertically transmitted obligate endosymbiont https://doi.org/10.1111/1462-2920.13887. It’s unlikely that it is transient or a contaminant.
Answer 4: We have added text noting that Torix-group Rickettsia in Culicoides are vertically transmitted obligate endosymbionts, unlikely to be transient or contaminant. Now, discussion section had been improved.
Reviewer 3 Report
Comments and Suggestions for Authors
Review
Title: Characterization of the Bacteriome of Culicoides reevesi from Chihuahua, Northern Mexico: Symbiotic and Pathogenic Associations
The paper examined the bacteria associated with Culicoides reevesi, a species found in northern Mexico by collecting groups of adult midges and analyzed their bacteria using genomic and bioinformatic methods. The paper is in general well presented, however some issues must be solved before it can be considered for publications.
Detailed comments
The abstract and the introduction is well presented, at the end if the Introduction, please provide a point by point hypothesis instead of this general presentation. As example, why is important to present the symbionts and the pathogens of Culicoides reevesi?
Methods: Please explain, why only one field was sampled, it looks as the whole 125 female were collected from the same region. Why?
What was the reason to collect 125 samples and not more or less?
How were the data used, did you created a database in OTU or how? How were the species identified, and what was the reason of presenting some of them at species, but others at family or Phyla level?
Results: The figure 2 is not appropriate if only one bar is presented. Please reconsider by using separate bars for each Phyla.
The presentation in a separate taxonomic level the pathogenic bacterial at species, and the symbionts at genus of family level does not make sense to me. What was the reason to do so. Please also explain the connections between these two groups in the results and discussions.
Please verify the resolution of the figure 3, is so different, please correct, especially at symbionts.
What is the reason of the results presented in figure 4 is presented separately, please give connections with the previous data and results, and please also mention in hypothesis.
Discussions must have a much stronger connection between the pathogens and symbionts community.
Author Response
Dear Reviewer,
All comments have been addressed, and they have significantly improved the quality and clarity of the manuscript. Thank you.
Author's Reply to the Review Report (Reviewer 3)
Detailed comments
Comment 1: The abstract and the introduction is well presented, at the end if the Introduction, please provide a point by point hypothesis instead of this general presentation. As example, why is important to present the symbionts and the pathogens of Culicoides reevesi?
Answer 1: Abstract and introduction are already improved.
Comment 2: Methods: Please explain, why only one field was sampled, it looks as the whole 125 female were collected from the same region. Why?
Answer 2: Thank you for the comment.
Indeed, the collection of Culicoides specimens was carried out at a single site. For this reason, the study was submitted as a Communication rather than a full research article (which, according to the journal Insects (MDPI) guidelines, are short articles presenting groundbreaking preliminary results or significant findings that form part of a larger, ongoing study).
However, it is an important contribution, since it is the first study in Mexico to evaluate the bacteriome of Culicoides. Future research will expand sampling to additional sites and seasons to further explore spatial and temporal variation in microbial composition.
Comment 3: What was the reason to collect 125 samples and not more or less?
Answer 3: The number of 125 C. reevesi females was selected to obtain a representative sample of the local population while ensuring sufficient material for DNA extraction and sequencing. The populations of Culicoides in this region are relatively low; however, the sampling site is of high economic importance, as it is a livestock-producing area where vector-borne diseases can have significant impacts. Therefore, understanding the bacteriome of C. reevesi from this area provides valuable baseline information for assessing its potential role in pathogen transmission and for guiding future surveillance and control strategies.
Comment 4: How were the data used, did you created a database in OTU or how?
Answer 4: The sequencing data were processed through the Galaxy EU platform using the QIIME2 pipeline. After quality filtering, reads were analyzed with the DADA2 algorithm to infer amplicon sequence variants, which provide higher taxonomic resolution than traditional OTUs. Then, taxonomy was performed using SILVA database. So, no separate OTU database was created; instead, ASVs were used throughout the study for all downstream analyses.
Comment 5: How were the species identified, and what was the reason of presenting some of them at species, but others at family or Phyla level?
Answer 5: Bacterial identification was performed using the QIIME2 classify-sklearn algorithm with the SILVA 138 database as a reference.
By other hand, the taxonomic performance depends on the degree of sequence conservation within the amplified 16S rRNA region and on the extensiveness of the reference database of the QIIME2 software. Thus, some bacterial reads could be positively assigned to the species level, while others could only be resolved to the family or phylum level due to limited reference sequences for certain taxa.
Comment 6: Results: The figure 2 is not appropriate if only one bar is presented. Please reconsider by using separate bars for each Phyla. The presentation in a separate taxonomic level the pathogenic bacterial at species, and the symbionts at genus of family level does not make sense to me. What was the reason to do so. Please also explain the connections between these two groups in the results and discussions.
Answer 6: Thank you for the comment. Pathogenic bacteria were presented at the species level because the identification platform (CZ ID) contains a larger number of reference sequences for pathogenic taxa, allowing higher-resolution classification. In contrast, symbiotic bacteria are less represented in public databases, which limits their identification to the genus or family level. This difference reflects current database coverage rather than analytical bias.
Now, the fig 2 was improved. Now, there is 2 plots. Plot A explain the global relative abundance of phyla across all pooled samples, whereas plot B shows the relative abundance of each pools.
Comment 7: Please verify the resolution of the figure 3, is so different, please correct, especially at symbionts.
Answer 7: Improved now. Reviewer 2 made a similar comment, and this figure has already been revised accordingly.
Comment 8: What is the reason of the results presented in figure 4 is presented separately, please give connections with the previous data and results, and please also mention in hypothesis.
Answer 8: Figure 2 was presented separately because it corresponds to a distinct analysis focused on functional pathway categories predicted by PICRUSt2, which are not based on taxonomic classification but on inferred metabolic functions of the bacterial community.
Round 2
Reviewer 1 Report
Comments and Suggestions for Authors
I believe the current manuscript is largely descriptive and relatively limited in scope. The study appears to lack sufficient innovation and does not yet meet the standards required for publication in the journal. Therefore, I would respectfully maintain my recommendation for rejection.
Author Response
Final comment: Thank you for your comments. As mentioned previously, this study is a communication. According to MDPI guidelines, it meets the criteria for preliminary studies.
Reviewer 3 Report
Comments and Suggestions for Authors
Agree with the corrections made by the authors.
Author Response
Final comment: Thanks for your feedback! We've implemented it, and the article has greatly improved.